# Significance of BK Polyomavirus in Long-Term Survivors after Adult Allogeneic Stem Cell Transplantation

**DOI:** 10.3390/biology10060553

**Published:** 2021-06-19

**Authors:** Thomas Neumann, Nandette Peters, Jennifer Kranz, Desiree L. Dräger, Florian H. Heidel, William Krüger, Laila Schneidewind

**Affiliations:** 1Department of Hematology/Oncology, University Medical Center Greifswald, 17475 Greifswald, Germany; thomas.neumann@med.uni-greifswald.de (T.N.); nandette.peters@stud.uni-greifswald.de (N.P.); Florian.heidel@uni-greifswald.de (F.H.H.); 2Department of Urology and Kidney Transplantation, Martin-Luther-University, 06120 Halle/Saale, Germany; jennifer.kranz@rwth-aachen.de; 3Department of Urology, St. Antonius Hospital gGmbH, 52249 Eschweiler, Germany; 4Department of Urology, University Medical Center Rostock, 18055 Rostock, Germany; desiree.draeger@med.uni-rostock.de

**Keywords:** allogeneic stem cell transplantation, BK polyomavirus (BKPyV), chronic kidney disease

## Abstract

**Simple Summary:**

Allogeneic stem cell transplantation is a curative treatment option for several hematological diseases. Data about health status and late complications of long-term survivors of this therapy are limited, so we conducted a prospective study. This analysis focusses on kidney function and urological complications. Interestingly, the BK polyomavirus plays an important role in this patient population and can lead to severe impairment of kidney function. This was only previously described in the acute situation following transplantation. Further studies should address causal therapy development for this severe viral infection.

**Abstract:**

Background: Allogeneic stem cell transplantation (aSCT) is a common treatment for a variety of hematological diseases. Advances in transplantation practices have led to an increasing number of long-term aSCT survivors, but data about health status and late complications are sparse. This analysis focusses on kidney function and urological complications in this population. Methods: This study is a prospective unicentric non-interventional trial. Before starting the study, we obtained the approval of the local ethics review board. Furthermore, the study was registered at WHO Clinical Trial Registry. The study protocol is available via UTN. Results: We were able to include 33 patients with a mean age of 60.5 years (SD 11.1). The median survival time following allogeneic stem cell transplantation was 9.0 years (IQR 8.5–13.0). Five patients (15.2%) had BKPyV viruria with mean 218.3 (SD 674.2) copies/mL. BKPyV viruria was significantly linked to pre-existing chronic kidney failure (*p* = 0.019), creatine > 100 µmol/L (*p* < 0.001), and cystatin c > 1.11 mg/L (*p* = 0.021), respectively. We were not able to identify a single risk factor for BKPyV viruria in univariate or multivariate Cox regression. Conclusions: BKPyV-associated nephropathy might be one reason for impaired kidney function in long-term survivors of aSCT.

## 1. Introduction

Allogeneic stem cell transplantation (aSCT) is a common treatment for a variety of hematological diseases. Advances in transplantation practices and supportive care have led to improved outcomes and an increasing number of long-term aSCT survivors. Most deaths after aSCT occur within the first 2 years as a result of relapse, acute or chronic Graft-versus-host disease (GvHD), infections, or other acute or subacute toxicities [1]. However, patients who survive beyond 2 years after aSCT also have an increased risk of long-term complications, which may impact on their survival and quality of life [2]. Furthermore, chronic kidney disease (CKD) is an important late morbidity among the long-term survivors of aSCT [3,4,5,6,7,8]. Reported risk factors for CKD after aSCT include older age at the time of transplantation, exposure to total body irradiation as part of the conditioning regimen, chronic GvHD, and post-transplant event of acute kidney injury [3,7,9]. Unfortunately, the knowledge about chronic kidney disease, urological complications, and infections is very limited, especially in adult aSCT and in very long-term survivors (more than 5 years after aSCT) [3].

One specific acute complication of aSCT is the BK polyomavirus (BKPyV)-associated hemorrhagic cystitis, which can occur in 5–60% of cases. Likewise, BKPyV-associated nephropathy (BKVAN) is a major challenge in the management of aSCT patients and can lead to severe morbidity and even mortality [10,11,12,13,14]. Other viruses associated with urological complications following aSCT are JC polyomavirus (JCPyV) and uro-pathogenic adenoviruses [10,15,16,17]. To the best of our knowledge, data concerning the relevance of these viruses in long-term survivors of aSCT are not available.

Consequently, we conducted a prospective clinical study to assess the health status and quality of life in long-term survivors of adult aSCT (>5 years following transplantation) at our institution. Regarding the primary end point, this descriptive analysis focuses on urological complications, kidney function, and viral infections. Secondary endpoints included association of urological complications, kidney function. and BKPyV, with the main clinical parameters and identification of risk factors for BKPyV-associated diseases.

## 2. Materials and Methods

### 2.1. Development of the Study and Study Population

The study was designed according to the guidelines in the synthesis of qualitative research (ENTREQ) found on the equatornetwork.org [18]. Before starting the study, we obtained the approval of the local ethics review board at the University Medicine in Greifswald (BB 146/15 from 20 October 2015). Furthermore, the study was registered at the WHO Clinical Trial Registry (Universal Trial Number UTN U1111-1176-5256). Formally, this study is a prospective unicentric non-interventional trial. The inclusion criteria were adult patients over 18 years receiving their first allogeneic stem cell transplantation at the University Medical Center Greifswald for underlying hematological disease, after at least 5 years of survival following stem cell transplantation and no clinical signs of relapse or progress of underlying hematological disease. If the inclusion criteria were met and an informed consent was obtained from the patient, there were no further exclusion criteria. All relevant patient data were collected according to the study protocol, which can be assessed with the Universal Trial number (UTN). From April 2019 until August 2020, we were able to include 33 patients in our study. Three patients from our center, who met the inclusion criteria, declined to participate in the study.

### 2.2. Definitions and Statistical Analysis

The term “viruria” was defined as positive quantitative polymerase chain reaction (PCR) for the viruses BKPyV, JCPyV (LightMix Polyoma JC-BK, manufacturer Tib Molbiol, Berlin, Germany), and adenovirus (LightMix adenoviruses, manufacturer Tib Molbiol, Berlin, Germany) in the urine. Bacterial urinary tract infection (UTI) was defined as urogenital symptoms with positive urine culture and more than 10^5^ bacterial colony forming units (CFU). Additionally, we defined cytomegalovirus (CMV) reactivation as a reactivation requiring treatment according to international standards. Chronic kidney failure was defined according to the KDIGO clinical practice guidelines (Kidney Disease Improving Global Outcome), and acute renal failure was defined according to the definition of the Acute Kidney Injury Network (AKIN).

For each numeric variable, the numeric distribution was preliminarily assessed by the Kolmogorov–Smirnov test. Descriptive statistics were performed with mean and standard deviation for normal distribution or with median and IQR for non-parametric data. For parametric continuous variables, the Student t-test was used, and for parametric categorical variables, the chi-square test or the Fisher exact test was used. For risk factor assessment univariate and multivariate, the Cox regression method was used, and the significance was tested with the Wald statistic. Kaplan–Meier plots were used to estimate the median overall survival. All reported *p*-values were based on a two-sided hypothesis; *p* < 0.05 was considered to be significant. All statistical calculations were performed using a statistical package for the Social Sciences 26.0 software (SPSS Inc., Chicago, IL, USA).

## 3. Results

### 3.1. Demographic Characterization of the Study Population

We included 33 patients with a mean age of 60.5 years (SD 11.1). Another three patients who met the inclusion criteria declined to participate in this study (2 males; 1 female). Acute myeloid leukemia (AML) was the most frequent underlying hematological disease (36.4%). The median survival time following allogemic stem cell transplantation was 9.0 years (IQR 8.5–13.0). Pre-existing chronic kidney failure was seen in two patients (6.1%), while preexisting urological disease was quite frequent, e.g., 21.2% urolithiasis in the patient history. Table 1 gives an overview of the demographic characteristics of the study population.

### 3.2. Urological Infections and Kidney Function

At the time of presentation for our study, 6 patients (18.2%) had abnormal high creatine and 10 patients had abnormal high cystatin C (30.3%). Postrenal genesis of kidney failure was ruled out by ultrasound in these patients. Furthermore, 3 patients (9.1%) had significant asymptomatic bacteriuria all with E. coli and 5 patients (15.2%) had BKPyV viruria with mean 218.3 (SD 674.2) copies/mL. These findings were difficult to predict since there was no significant association with microhematuria or leucocyturia and no patient reported urinary tract symptoms. Interestingly, BKPyV viruria was significantly linked to preexisting chronic kidney failure (*p* = 0.019), creatine > 100 µmol/L (*p* < 0.001), and cystatin c > 1.11 mg/L (*p* = 0.021), respectively. Table 2 gives an overview about urological infections, kidney function, and their associations with BKPyV viruria, while Figure 1 illustrates the association of creatine, BKPyV copies in urine, and chronic kidney failure.

### 3.3. Association of BKPyV Viruria with Immunological Factors

The presence of BKPyV in urine was very difficult to predict, since we were not able to identify a significant association with any immunological factor and BKPyV viruria, such as leukopenia, shortage of immunoglobulins, or GvHD. Table 3 shows in detail the associations of BKPyV with immunological factors. Furthermore, we were not able to identify a single risk factor for BKPyV viruria in univariate or multivariate Cox regression. However, we visualized the lymphocyte subtyping of all five patients with BKPyV viruria in Table 4. All five patients had a relative or absolute shortage of T4 helper cells (three patients with relative shortage, two patients with absolute shortage, respectively).

## 4. Discussion

We conducted a prospective single-institutional non-interventional clinical study and sought to analyze health status of long-term (>5 years) survivors of adult aSCT. This was a subgroup-analysis focusing on urological complications, kidney function, and urological infections. Regarding the study population, we reported on an older patient cohort with a mean age of 60.5 years. As expected, the most frequent underlying disease was AML. Otherwise, the study population appeared heterogenous, which is also expected in studies analyzing patients undergoing adult aSCT for various diseases [1]. Of note, follow-up of our patient cohort was long (median survival 9.0 years). Most analyses published so far, that report on aSCT long-term survivors include patients who survived 2 years after transplantation. In contrast, our study provides first evidence for long-term complications. In their landmark study of 2-year survivors of aSCT, Wingard et al. concluded that the prospect for long-term survival was excellent for 2-year survivors of aSCT. However, life-expectancy remains lower than expected. Performance of aSCT earlier in the course of disease, control of GvHD, and enhancement of immune reconstruction, less toxic regimens, and prevention, as well as early treatment of late complications, are needed [1]. Consequently, a rigorous follow-up to trigger early intervention and prevent late complications is necessary, and further studies in long-term survivors of aSCT are clearly warranted. Notably, Wingard et al. did not focus on kidney function. To our knowledge, Jo et al. conducted the only analysis available in the literature on kidney function in 5-year survivors of aSCT. This was a retrospective single-institutional case-series including 106 patients. The authors concluded that careful monitoring of renal function is required for long-term survivors after aSCT, especially in patients who experienced acute kidney injury and in older patients. Furthermore, they reported that patients with CKD had a lower overall survival rate (HR 4.11; 95% CI 1.3–13.0) than patients without CKD and the cumulative incidence of CKD was 25.1% over 10 years [3]. However, they did not analyze urological complications or infections. This is of major interest, since BKPyV could induce BKPyV-associated nephropathy with kidney failure. Taken together, the relevance of BKPyV in long-term survivors of aSCT was never studied before.

Our results regarding CKD and renal failure are comparable to the study from Jo et al. [3]. At the time of presentation of the study, there were no urological infections or other complications, while pre-existing urological diseases were frequently detectable, except from asymptomatic bacteriuria and BKPyV viruria. Surprisingly, BKPyV viruria was significantly associated with pre-existing CKD and abnormal high kidney values (creatine, cystatin C). This was never described before, and we must assume that BKVAN also plays a significant role in long-term survivors of aSCT, while the other important BKPyV-associated disease, hemorrhagic cystitis, was not detected in this population. However, it is still difficult to predict the presence of BKPyV viruria, especially when not associated with microhematuria. Moreover, association of BKPyV viruria with immunological factors remained a challenge, and specific risk factors for BKPyV viruria in univariate and multivariate Cox regression could not be identified. One interesting aspect regarding failed virus control is that all five patients diagnosed with BKPyV viruria showed relative or absolute shortage of T4 helper cells. Further evaluation of the phenomenon is clearly warranted, since virus-specific T cells can be used to monitor and treat BKVAN, as shown for patients undergoing kidney transplantation [19,20,21,22].

Consequently, this raises the question for BKVAN therapy. So far, no causal treatment has been established [13]. Several compounds with activity against BKPyV have been described. Here, cidofovir was studied extensively; however, it is very nephrotoxic itself. Therefore, application in patients who already suffer from kidney failure is not reasonable [23]. Our results suggest that targeting the immune response to BKPyV might be a promising approach for further studies. Recently, approaches for experimental evaluation in long-term survivors of aSCT have been described. Our group reported on interleukin 11 (IL-11), which may represent a promising therapeutic target to treat BKPyV infections [24]. So far, careful monitoring of kidney function and BKPyV viral load remain the standard of care in the long-term follow up of aSCT patients.

Despite the prospective nature and novelty regarding BKPyV in long-term survivors of aSCT, we must assume that our study also has several limitations, for example, exposure to nephrotoxic substances and plasma viral loads of BKPyV have not been assessed.

## 5. Conclusions

CKD is a clinical challenge in long-term survivors of aSCT. BKVAN may contribute to CKD in this population. Here, we present the first study investigating this phenomenon. BKVAN is difficult to predict, and no therapy is established. Further studies in long-term survivors of aSCT should focus on careful monitoring of kidney function, BKPyV viruria, plasma viral load, and therapy development by targeting the immune response.

## Figures and Tables

**Figure 1 biology-10-00553-f001:**
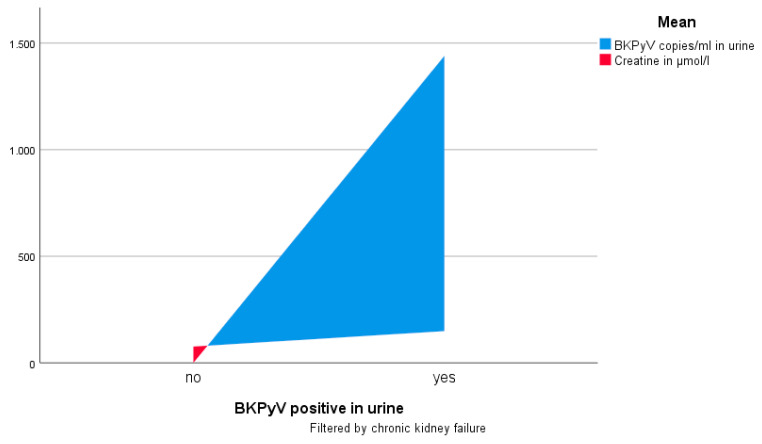
Association of creatine, BKPyV copies in urine, and chronic kidney failure.

**Table 1 biology-10-00553-t001:** Demographic characterization of the study population (n = 33).

**Sex**	**Female 13 (39.4%)**	-	-
	**Male 20 (60.6%)**		
**Age**	-	60.5 (11.1)	-
**Underlying hematological disease**	AML 12 (36.4%)	**-**	**-**
	NHL 11 (33.3%)		
	MPS 4 (12.1%)		
	ALL 2 (6.1%)		
	MDS 2 (6.1%)		
	MM 2 (6.1%)		
**Survival in years following transplantation**	**-**	**-**	9.00 (8.5–13.0)
**Donor**	Related 9 (27.3%)	**-**	**-**
	Matched-unrelated 15 (45.5%)		
	Mismatched-unrelated 9 (27.3%)		
**Number of mismatches**	**-**	**-**	0 (0.0–0.5)
**Donor chimerism > 95%**	33 (100.0%)	**-**	**-**
**BMI**	**-**	28.4 (4.9)	**-**
**ECOG Performance Status**	0-> 21 (63.6%)	**-**	**-**
	1-> 10 (30.3%)		
	2-> 2 (6.1%)		
**Preexisting chronic kidney failure**	KDIGO II 1 (3.0%)	**-**	**-**
	KDIGO III 1 (3.0%)		
**Preexisting urological disease**	Urolithiasis 7 (21.2%)	**-**	**-**
	Voiding dysfunction 5 (15.2%)		
	Phimosis 1 (3.0%)		
	Varicocele 1 (3.0%)		
	NMBC 1 (3.0%)		
	VUR 1 (3.0%)		
	BPH 1 (3.0%)		

AML = acute myeloid leukemia; NHL = non-Hodgkin lymphoma; MPS = myeloid proliferative syndrome; ALL = acute lymphatic; leukemia; MDS = myeloid dysplastic syndrome; MM = multiple myeloma; BMI = Body mass index; ECOG = Eastern Cooperative Oncology Group; KDIGO = kidney disease improving global outcome; NMBC = non-muscle invasive bladder cancer; VUR = vesicoureteral reflux; BPH = benign prostatic hyperplasia.

**Table 2 biology-10-00553-t002:** Urogenital infections, kidney function, and association with BKPyV viruria (n = 33).

Parameter	N (%)	Mean (SD)	Median (IQR)	Association with BKPyV Viruria *p*-Value
**BKPyV viruria**	5 (15.2%)	Mean copies/mL 218.3 (674.2)	**-**	**-**
**Preexisting chronic kidney failure**	2 (6.1%)	**-**	**-**	*p* = 0.019
**Creatine > 100 µmoL/L**	6 (18.2%)	**-**	**-**	*p* < 0.0001
**Cystatin C > 1.11 mg/L**	10 (30.3%)	**-**	**-**	*p* = 0.021
**Microhematuria**	0 (0.0%)	**-**	**-**	**-**
**Leukocyturia**	9 (27.3%)	**-**	**-**	*p* = 0.597
**Significant positive bacterial urine culture (10^5^ CFU)**	3 (9.1%) All E. coli	**-**	**-**	*p* = 0.400
**Preexisting urolithiasis**	7 (21.2%)	**-**	**-**	*p* = 0.282
**Preexisting voiding dysfunction**	5 (15.2%)	**-**	**-**	*p* = 0.569

BKPyV = BK polyomavirus; CFU = colony forming units.

**Table 3 biology-10-00553-t003:** Association of BKPyV viruria with immunological factors (n = 33).

Parameter	N (%)	Mean (SD) of Whole Population	Median (IQR) of Whole Population	Association with BKPyV Viruria *p*-Value
**Leukopenia (<4.3 Gpt/L)**	5 (15.2%)	6.0 (1.6)	-	*p* = 1.000
**Lymphopenia (<1.0 Gpt/L)**	3 (9.1%)	2.1 (0.9)	-	*p* = 0.400
**CR*P* > 5.0 mg/dL**	8 (24.2%)	2.5 (3.7)	-	*p* = 0.302
**Shortage of Immunoglobulin G (<7.0 g/L)**	4 (12.1)	2.2 (4.3)	-	*p* = 0.500
**Shortage of Immunoglobulin A (<0.7 g/L)**	2 (6.1%)	-	2.2 (1.8–2.8)	*p* = 1.000
**Shortage of Immunoglobulin M (<0.4 g/L)**	2 (6.1%)	1.0 (0.6)	-	*p* = 1.000
**More than 4 infections/year**	2 (6.1%)	-	-	*p* = 1.000
**CMV reactivation post-transplant**	4 (12.1%)	-	-	*p* = 0.571
**EBV/PTLD post-transplant**	0 (0.0%)	-	-	-
**Toxoplasmosis post-transplant**	1 (3.0%)	-	-	*p* = 1.000
**Acute GvHD post-transplant**	14 (42.4%)	-	-	*p* = 0.628
**Chronic GvHD post-transplant**	6 (18.2%)	-	-	*p* = 0.216
**Still existing chronic GvHD**	6 (18.2%)	-	-	*p* = 0.216
**Mismatched Donor**	8 (24.2%)	-	-	*p* = 0.574

BKPyV = BK polyomavirus; CRP = C-reactive protein; CMV = cytomegalovirus; EBV = Epstein-Barr virus; PTLD = post-transplant; lymphoproliferative disease; GvHD = Graft-versus-host disease.

**Table 4 biology-10-00553-t004:** Subtyping of lymphocytes with fluorescence-activated cell sorting of BKPyV viruria patients (n = 5).

Parameter	Patient 1		Patient 2		Patient 3		Patient 4		Patient 5		Reference
	%	/µL	%	/µL	%	/µL	%	/µL	%	/µL	/µL
**T cells**	62.96	1146	44.88+	1176	77.47	1193	53.60−	525−	71.42	1321	690–2450
**Ga/de T cells (indirect)**	1.87	34	8.13+	213+	4.11	63	6.69+	66	4.60	85	<5%
**T4 cells**	26.47−	482	20.47−	536	31.05−	478	26.14−	256−	21.08−	390−	410–1590
**T8 cells**	35.16	640	16.80−	440	42.97+	662	21.13	207	46.79−	866	190–1140
**NK cells**	23.70	431	16.44	431	9.93	153	18.47	181	12.02	222	90–590
**B cells**	13.25	241	38.48+	1008+	12.20	188	27.51+	270	16.49	305	90–660
**CD3+ HLA-DR+**	9.42	171	8.06	211	12.59	194	4.39	43	13.06+	242+	<230
**CD3+ CD57+**	10.88+	198	12.91+	338+	16.69+	257+	14.17+	139	31.83+	589+	2–10%
**T4/T8 ratio**	0.75	-	1.22	-	0.72	-	1.24	-	0.45−	-	0.6–2.8
**Monocytes**	Normal CD13 cluster		Normal CD13 cluster		Normal CD13 cluster		Normal CD13 cluster		Normal CD13 cluster		-

## Data Availability

All data are available upon request from the corresponding author.

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
