# Peer review of "Significance of BK Polyomavirus in Long-Term Survivors after Adult Allogeneic Stem Cell Transplantation"

_biology, 2021, doi:10.3390/biology10060553_

Round 1
Reviewer 1 Report
In this prospective study, Neumann and colleagues focus on kidney function and urological complications of 33 long-term survivors (mean: 9.0 years, IQR: 8.5-13.0) from adult allogeneic stem cell transplantation, a therapeutic treatment option for several hematological diseases. The Authors concluded that BK Polyomavirus (BKV) plays an important role in these patients’ population, and it significantly correlates with well know parameters of kidney failure. It is also well known the BKV association with nephropathies after kidney transplantation and hemorrhagic cystitis after bone marrow transplant. However, the data reported in this paper needs additional revisions in order to corroborate the main finding of this study: the role of BKV in long-term survivors from adult allogeneic stem cell transplantation. In my opinion, the Authors should consider the hereby list of the majors' concerns about this manuscript.
- The WHO Clinical Trial Registry number should be reported in Materials and Methods or better in a dedicated section, not in the abstract.
- The introduction should be improved adding the state of BKV and its association with kidney failure and urological conditions.
- The sentence states that three patients who declined to participate in the study should be reported as results and not in the Materials and Methods section. I would also not include these patients in the study population if they did not affect the results.
- In the Materials and Methods section, the PCR assays used to detect BKV, JCV, and Adeno in the urine should be described and the specificity and sensitivity of the assay should be reported. Moreover, the results obtained regarding the viruses JCV and Adeno are not mentioned in the results. Is there any reason for this choice?
- On page 3 line 116, I assume that Table 1, not Figure 1, gives an overview of the demographic characteristics of the study population.
- Figure 1 is not clear. I would provide a better description of the figure as a legend to figure 1 and I would suggest using another graphic method to show the data.
- I suggest adding a brief description of ECOG and KDIGO criteria for a better understanding of the reported data.
- On page 5 lines 132-133, the Authors state that five patients had BKV viruria with a mean of 218.3 copies/mL and I standard deviation value bigger than the mean (674.2 copies/mL). These data are mathematically irrelevant, and the result obtained should be reconsidered as a troubleshoot of the PCR assay instead. The BKV viruria strictly correlates with the specific immune asset of the patients at the time of the PCR assay, and usually, in a BKV diagnostic laboratory, a result as the one reported in this study should be reconsidered or retested considering the immune condition of the patients.
- The whole discussion section should be rewritten. In particular, the sentences reported at lines 167-170 and 176-178 are extracts of the abstracts from the paper listed in the references as 1 and 3, respectively. This trend is strongly not recommended because it could be in compliance with plagiarism rules. Therefore, I suggest the Author rephrasing these sentences in their own words.
- Finally, I would suggest the author revise the references list: 5 of the 24 cited papers are from the Authors of this manuscript.
Sincerely,
Author Response
Dear Reviewer,
Thank you for your valuable time and your thoughtful comments to improve our manuscript and even our research. Please, find our comments to your major concerns below and we have marked all changes in the manuscript in red color.
Major 1: Thank you! We have added the information to the material and methods part and deleted it from the abstract.
Major 2: We have improved our introduction n to make clearer statements about BKPyV and kidney function.
Major 3: We did not include these patients in our analysis, which is clear when we report the percentages – it is only to state how big our population at our institution was.
Major 4: We included the information about the PCR. We cannot report JCPyV and adenovirus since they were not detected, which is stated in the results.
Major 5: We have corrected this!
Major 6: The figure has a legend. We added information
Major 7: ECOG and KDIGO is used in every day practice, so we must say that every physician working on patient need to know them.
Major 8: We a very sorry, because we think that these data are relevant. Quantitative PCR is a very sensitive method, as you know, and these data show that even a small number of copies of BKPyV might trigger nephropathy!
Major 9: We have improved our methods section, but we can certainly not agree with you that this is plagiarism – since we also have written the abstract ourselves!
Major 10: We agree with you that certainly it is not good to only cite yourself. Unfortunately, we are one of the only research groups who performed prospective clinical trial on that topic!
Thanks again for your help and very Kind Regards,
Laila Schneidewind
Reviewer 2 Report
The manuscript is well-written and the conclusion is straightforward.
There is one important statement that needs to be clarified by the authors. The authors claim that this is a long-term prospective trial, however, the described methods do not entirely reflect the claimed nature of the study. Some suggestions can be considered as follow:
- 33 patients were involved. What was the follow-up time of these patients?
- What was the trend in kidney function in the follow-up period?
- What was the trend in BK viruria in the follow-up period?
- The data will be more convincing if provided longitudinally.
Author Response
Dear Reviewer,
Thank you for your valuable time and your thoughtful comments to improve our manuscript and even our research. Please, find our comments to your major concerns below and we have marked all changes in the manuscript in red color.
Major: We do not have a follow up time, we included patients surving longer than 5 years at our institution, so we state the median survival time in the manuscript! That why we can not provide the data longitudinally. What do you mean be trend? Median copy numbers and kidney function are stated in the manuscript, but we have over written our discussion, so that your requested information is more obvious to the reader.
Thanks again and very Kind Regards,
Laila Schneidewind
Round 2
Reviewer 1 Report
The authors did a good job by answering the concerns of this reviewer.